# A Chemerin Peptide Analog Stimulates Tumor Growth in Two Xenograft Mouse Models of Human Colorectal Carcinoma

**DOI:** 10.3390/cancers14010125

**Published:** 2021-12-28

**Authors:** Justa Friebus-Kardash, Petra Schulz, Sandy Reinicke, Cordula Karthaus, Quirino Schefer, Sebastian Bandholtz, Carsten Grötzinger

**Affiliations:** 1Department of Hepatology and Gastroenterology, Charité—University Medicine Berlin, Corporate Member of Free University Berlin and Humboldt University Berlin, 13353 Berlin, Germany; justa.friebus-kardash@uk-essen.de (J.F.-K.); p.schulz@metriopharm.com (P.S.); sandy.reinicke@bayer.com (S.R.); cordula.karthaus@uni-oldenburg.de (C.K.); quirino.schefer@regenold.com (Q.S.); sbandholtz@gmail.com (S.B.); 2Department of Nephrology, University Hospital Essen, University Duisburg-Essen, 45127 Essen, Germany; 3Partner Site Berlin, German Cancer Consortium (DKTK), 13353 Berlin, Germany; 4German Cancer Research Center (DKFZ), 69120 Heidelberg, Germany

**Keywords:** chemerin, peptide analog, CG34, CMKLR1, GPR1, CCLR2, colorectal cancer, xenograft mouse model, tumor treatment

## Abstract

**Simple Summary:**

The chemoattractant adipokine chemerin has been found to be elevated in several types of cancer, including colorectal carcinoma. The functional role of chemerin in colorectal carcinoma, however, has not been elucidated to date. This study analyses the impact of the chemerin analog CG34 on proliferation, colony formation, and migration in the human colorectal cancer cell lines HCT116, HT29 and SW620. In addition, the effect of systemic CG34 treatment is investigated in two xenograft mouse models of colorectal cancer (HCT116-luc and HT29-luc). The results of this study suggest there is a stimulatory role of chemerin receptor activation on the growth of colorectal carcinoma.

**Abstract:**

Background: Chemerin plasma concentration has been reported to be positively correlated with the risk of colorectal cancer. However, the potential regulation of CRC tumorigenesis and progression has not yet been investigated in an experimental setting. This study addresses this hypothesis by investigating proliferation, colony formation, and migration of CRC cell lines in vitro as well as in animal models. Methods: In vitro, microscopic assays to study proliferation, as well as a scratch-wound assay for migration monitoring, were applied using the human CRC cell lines HCT116, HT29, and SW620 under the influence of the chemerin analog CG34. The animal study investigated HCT116-luc and HT29-luc subcutaneous tumor size and bioluminescence during treatment with CG34 versus control, followed by an ex-vivo analysis of vessel density and mitotic activity. Results: While the proliferation of the three CRC cell lines in monolayers was not clearly stimulated by CG34, the chemerin analog promoted colony formation in three-dimensional aggregates. An effect on cell migration was not observed. In the treatment study, CG34 significantly stimulated both growth and bioluminescence signals of HCT116-luc and HT29-luc xenografts. Conclusions: The results of this study represent the first indication of a tumor growth-stimulating effect of chemerin signaling in CRC.

## 1. Introduction

Chemerin is a regulatory adipokine involved in energy metabolism, regulation of immune function, and other physiological processes such as chemotaxis, differentiation, and proliferation. It exerts its action via its G protein-coupled receptors chemokine-like receptor 1 (CMKLR1), G protein-coupled receptor 1 (GPR1), and C-C chemokine receptor-like 2 CCRL2 [1,2]. Both clinical, as well as animal studies, have clearly demonstrated that secretion and systemic levels of chemerin rise with overweight and adiposity, and decline after diet, exercise-based weight loss, and bariatric surgery [3,4,5,6]. Chemerin has been shown to promote the chemotaxis of macrophages and NK cells and to enhance the migration of plasmacytoid dendritic cells [7,8,9]. As a chemoattractant in inflammation, chemerin was reported to exert action on NF-κB signaling, thereby promoting pro-inflammatory, but also anti-inflammatory activities [10,11,12]. In hypertension, chemerin protein levels are elevated, and experimental studies support the role of chemerin in the control of blood pressure, while inhibition of angiotensin-I-converting enzyme lowers chemerin serum levels [13,14,15,16,17].

Chemerin has also been associated with several functions of tumor progression, such as growth, invasion, metastasis, recruitment of immunocytes, and formation of new vessels [18,19]. In a seminal paper, Pachynski et al. found chemerin to suppress melanoma by recruiting natural killer cells [20]. In a chemically induced model of skin cancer, the expression of chemerin by keratinocytes has been demonstrated to block the distal phase of tumorigenesis [21]. Similarly, chemerin was found to counteract tumor progression in hepatocellular carcinoma (HCC) by suppressing IL-6 and GM-CSF expression. Further, it inhibits metastasis in HCC along a signaling pathway via CMKLR1, PTEN, and Akt [22,23]. Among patients with HCC, diminished chemerin biosynthesis has been attributed to poor prognosis as well as lowered infiltration of NK cells and dendritic cells [24]. Notwithstanding antitumor action by chemerin has been reported in a number of cancer types, there are also reports about tumor-enhancing functions. In neuroblastoma cells, inhibition of chemerin-induced CMKLR1 signaling was found to reduce clonal growth and cellular viability in vitro and to inhibit tumor growth in animal experiments [25]. Likewise, chemerin signals via CMKLR1 and GPR1 to enhance migratory activity and invasion of gastric cancer cells [26].

Regarding colorectal carcinoma (CRC), chemerin plasma concentration was found to be associated with the risk of incident CRC and to be independent of established CRC risk factors [27,28]. In addition, serum chemerin levels turned out to be positively associated with the presence of colorectal adenoma in men [29]. Furthermore, chemerin receptor CMKLR1 was detected at higher abundance in CRC tumor tissues than in margins [30].

So far, however, the potential regulation of CRC tumorigenesis and progression has not been investigated in an experimental setting. Recently, our group has reported the development of the metabolically stable high-affinity chemerin peptide agonist CG34 that potently activates both CMKLR1 as well as GPR1. Chelator-linked and radionuclide-loaded CG34 was able to detect via PET/MRI imaging breast cancer in a mouse model that used xenografts of the endogenously CMKLR1-expressing human cell line DU4475 [31]. 

In this study, the chemerin analog CG34 was used to investigate the possible regulation of three human CRC cell line models in vitro with regard to proliferation, colony formation, and migration. In addition, CG34 was utilized to study the influence of chemerin receptor signaling on growth and vascularization in two human CRC cell xenograft mouse models. The results of this study represent the first indication of a tumor growth-stimulating effect of chemerin signaling in CRC.

## 2. Materials and Methods

### 2.1. Reagents and Cell Culture

The luciferase-expressing cell lines HCT116-luc and HT29-luc were purchased from Caliper (Hopkinton, MA, USA). The cell line SW620-luc cell line was created by transduction in our laboratory. All other cell lines were purchased from ATCC/LGC Standards (Wesel, Germany) or CLS (Eppenheim, Germany). The cell lines were cultured in RPMI 1640 medium (Biochrom AG, Berlin, Germany) or McCoy’s 5A modified medium (Biochrom AG, Berlin, Germany), each supplemented with 10% fetal calf serum (Biochrom AG, Berlin, Germany). All tumor cell lines were cultured in an incubator (Labotect, Rosdorf, Germany) providing a humidified atmosphere at 37 °C with 5% CO_2_. Cells were passaged every 2–3 days.

### 2.2. RNA Isolation, Transcription, and RT-qPCR

The preparation of RNA for RT-qPCR experiments was performed according to the manufacturer’s instructions using the RNeasy Plus Mini Kit (Qiagen, Hilden, Germany). Isolated RNA was transcribed into cDNA using the High Capacity cDNA Reverse Transcription Kit (Applied Biosystems, Darmstadt, Germany). For this, 3.2 µg of RNA was transcribed into 80 µL of cDNA (final concentration: 40 ng/µL) according to the manufacturer’s protocol. In addition, as a negative control for each sample, a preparation without reverse transcriptase was analyzed in RT-qPCR. Quantitative real-time human-specific primer sets for TaqMan and CyberGreen amplifications were obtained from Life Technologies (Darmstadt, Germany), SIGMA (Deisenhofen, Germany), or Roche (Mannheim, Germany). Primer sequences, or product numbers, are provided in Table 1. The Fast Start TaqMan^®^ Probe Master Kit (Roche) additionally contains Taq DNA Polymerase, reaction buffer, and nucleotides (dATP, dCTP, dGTP, dUTP). For CyberGreen-based PCR, Sso Fast EvaGreen Supermix (Bio-Rad, Düsseldorf, Germany) was used. PCR was performed using the CFX96 real-time system (Bio-Rad). For this purpose, 30 ng of cDNA were added. The analysis was performed using qBase PLUS (Biogazelle, Zwijnaarde, Belgium) according to the ∆∆Ct-method. As multiple reference genes were included in the normalization, the geometric mean of the relative housekeeping expression was first determined and then this value was used to normalize the target gene expression. Subsequently, these data were normalized to a cq value of 34. 

### 2.3. Chemerin ELISA

For the detection of chemerin in cell culture supernatants, ELISA kits from DRG Instruments (Marburg, Germany) were used. Cells were incubated in a full medium at a density of approximately 70% for 24 h; supernatants were immediately used for ELISA or frozen at −80 °C. The ELISA was performed according to the manufacturer’s protocol with the supplied standards. MaxiSorp 96-well plates (Nunc, Thermo Fisher Scientific, Dreieich, Germany) were used. The measurement was performed in the SpectramaxPlus384 reader (Molecular Devices, Biberach an der Riß, Germany) at 650 nm (before the addition of the stop solution) or 450 nm (after addition of the stop solution) and the concentration was determined based on the standard curve using SoftMax Pro 5.3.

### 2.4. Proliferation Assay Using DAPI Staining of Nuclei

After reaching a confluence level of 60–70%, cells were trypsinized (trypsin/EDTA solution, Biochrom AG, Berlin, Germany) and seeded onto a 96-well plate with transparent, microscopy grade bottom (Becton Dickinson, Heidelberg, Germany). Cell seeding was then performed by pipetting 100 µL containing 10,000 cells per well. 

At 1 h after cell seeding, the substances to be tested, including negative controls (medium containing 1.6%, 0.8% or 0.6% FCS for HCT116-luc, HT29-luc, and SW620-luc, respectively, and positive controls (medium containing 10% FCS) were added to the cells in a volume of 100 µL per well. The chemerin analog CG34 (final concentration 10 µM) was diluted in the medium corresponding to the negative controls, as stated above. To avoid loss of substance activity over time, a medium with or without compound was exchanged every 24 h. 

After 96 h, cells were fixed for 10 min in a PBS-buffered 4% PFA solution (Herbeta Arzneimittel, Berlin, Germany) pipetted directly into the medium. Then, cells were subjected to 10 min of incubation with 5 µg/mL DAPI (Sigma-Aldrich, Munich, Germany) and 0.1% Triton (Merck, Darmstadt, Germany) in PBS. Plates were washed 2 times with PBS. After the last wash, PBS was left on the plate. The plate was covered with a transparent foil and stored at 4 °C until measurement. The number of nuclei was determined using high-content analysis with the IN Cell Analyzer automated microscope (GE Healthcare, Reading, UK). Data are reported as relative cell numbers in % of the positive control.

### 2.5. Proliferation Assay Using Bright-Field Microscopy of Growing Cells

After reaching a confluence level of 60–70%, cells were trypsinized and seeded onto a 96-well standard cell culture plate (Sarstedt, Nümbrecht, Germany). Cell seeding was then performed done by pipetting 100 µL each with 10,000 cells per well. Directly after cell seeding, 100 µL per well of RPMI1640 with 10% FCS containing the chemerin analog CG34 were added to the cells. As negative controls, other wells were incubated in RPMI1640 with 10% FCS only. Plates were then placed for up to five days into an automated microscopy system (Incucyte S3, Essen Bioscience, Göttingen, Germany). Bright-field images of each well were taken every two hours. To avoid loss of substance activity over time, a medium with or without compound was exchanged every 48 h. Using the Incucyte software, cell confluence was determined by an adapted segmentation algorithm. Using kinetic confluency data, the rate constant k for cell growth up to three days after seeding was calculated using the exponential growth equation (Y = Y0 × exp(k × X) in GraphPad Prism 7.0 (GraphPad Software, San Diego, CA, USA). 

### 2.6. Colony Formation Assay

Cells were seeded in a flask and incubated for 3 days until reaching 70% confluence. Cells were then trypsinized, suspended in serum-free RPMI1640 medium, and counted using a Neubauer counting chamber. Four ml of a cell suspension containing 30,000 cells per 1 mL of FCS-free medium were prepared. In parallel, a sterile β-mercaptoethanol solution was prepared (5 µL of 2-mercaptoethanol (Sigma-Aldrich, Munich, Germany)) in 8.45 mL of PBS. In addition, methylcellulose aliquots (Sigma-Aldrich, Munich, Germany) were pre-warmed at 37 °C. Under sterile conditions, a mixture of 3.6 mL of methylcellulose, 2.7 mL of FCS, 60 µL of mercaptoethanol solution, 770 µL of Iscove’s medium (Biochrom AG, Berlin, Germany), and 300 µL of the cell suspension was prepared in a tube. The tube was shaken vigorously for approximately 5 min until the preparation took on a pink color. Then, 1.63 mL of agar previously boiled in the microwave and dissolved in the serum-free medium were added to the batch, shaken vigorously repeatedly, and placed in the cell incubator for 20 min. After the incubation period had elapsed, 1 mL of the resulting cell mixture was spread evenly on a gridded 3.5-cm cell culture dish using an insulin syringe, taking care to prevent the formation of air bubbles. For both CG34 treatments as well as the medium control, three small cell culture dishes (2 mm grid, Nalge Nunc International, Rochester, NY, USA) were filled with the cell preparation and placed in a large cell culture dish, which was previously loaded with a small dish filled with PBS to prevent the preparation from drying out. The next day, each cell batch was treated with 100 µL of the substance to be tested for stimulation of colony formation. The negative control consisted of the equivalent treatment with 100 µL of medium containing a concentration of FCS specified for the cell line under investigation (see DAPI assay). CG34 was diluted (100 nM) in the medium of the FCS concentration corresponding to the negative control. Subsequently, compound or negative control additions were made once daily, with only 10 µL added to each batch. The large cell culture dishes containing the preparations were incubated in an incubator for 7 days. On the last day of the experiment, tumor cell colonies were manually counted at a defined size of at least 20 cells per colony under a bright-field microscope at 40× magnification. Data are reported as relative cell numbers in % of the negative control.

### 2.7. Migration Assay

After reaching a confluence level of 80–90%, cells were trypsinized and seeded in a 96-well standard cell culture plate (Sarstedt, Nümbrecht, Germany). Cell seeding was then performed done by pipetting 100 µL of RPMI1640 with 10% FCS with 75,000 cells per well. The next day, cell monolayers were treated with a 96-head scratch tool. Cells were washed once with a medium. Thereafter, 200 µL per well of RPMI1640 with 10% FCS containing the chemerin analog CG34 (10 µM) was added to the cells. As negative controls, other wells were incubated in RPMI1640 with 10% FCS only. Plates were then placed into an automated microscopy system (Incucyte S3, Essen Bioscience, Göttingen, Germany) for up to 24 h. Using the Incucyte software, the % relative change in wound density from 8 to 20 h after compound addition was determined by an adapted segmentation algorithm. 

### 2.8. Treatment Study

To analyze the potential tumor growth effect of CG34 in vivo, a treatment study was performed. Two luciferase-expressing xenograft mouse models for human colorectal carcinoma were used: HCT116-luc and HT29-luc. 12 female 8–10 weeks old NMRI nude mice were each implanted subcutaneously with 5 × 10^6^ cells in 100 µL RPMI1640 each of the HT29-luc (left flank) or HCT116-luc (right flank). Tumor growth was monitored once weekly from the first day of the experiment using both caliper size measurements and bioluminescence imaging (IVIS Lumina, Caliper [Hopkinton, MA, USA]). On the seventh day, the 12 animals were randomized into two groups with six animals per group. Treatment was started on day eight of the experiment and terminated on day 28. Treatment was administered in a blinded manner (the investigators did not know the assignment of groups until after evaluation), with one group receiving intraperitoneal injections of 200 µL of CG34 (50 nmol) per mouse once daily and the other group receiving equivalent treatment with 200 µL of water. After measuring the length and width of the tumors with a caliper, volume was calculated according to the formula v = 0.5 × (length × width^2^). 

For bioluminescence imaging, each mouse received an intraperitoneal weight-adapted injection of D-luciferin (150 µL) at a dose of 150 µg/g mouse 5–10 min before imaging. Mice were then placed under isoflurane anesthesia and transferred to the imager, where bioluminescence signal detection took place. The exposure time was 1 min at the beginning for small tumors and would be reduced to a minimum of 10 s as the tumor size increased during the experiment. The Living Image 3.1 software (Caliper) was used to count the emitted photons, ultimately determining the photon yield in a region of interest corresponding to the tumor under investigation. In addition, animal weight was monitored twice a week. At the designated experimental endpoint, day 31, animals were anesthetized with isoflurane and euthanized by cervical dislocation. The endpoint tumor volume was determined using a caliper of tumors on sacrificed mice (length, width, and height at autopsy). In addition, the weight of the tumors obtained at autopsy was determined. The excised tumors were frozen in liquid nitrogen and stored at −80 °C. 

### 2.9. CD31 Histology

To visualize tumor vascularization, immunohistochemical staining was performed using an antibody directed against the mouse endothelial cell surface molecule CD31 (JC70A, DAKO, Glostrup, Denmark). 10 µm-thick cryosections were fixed with 4% PFA solution for 20 min at room temperature. To block endogenous tissue peroxidases and prevent the nonspecific conversion of the dye AEC added later, sections were incubated with a 0.3% H_2_O_2_ solution for 10 min. Furthermore, an avidin-biotin blockade was performed using the avidin-biotin blocking kit (DAKO, Glostrup, Denmark). To eliminate nonspecific protein interactions with the primary antibody, sections were blocked with 2% skim milk for 30 min. Primary antibody was dissolved 1:50 in 0.1% BSA in PBS and incubated overnight at 4 °C in a humid chamber. As a negative control, the sections were incubated with only the 0.1% BSA solution instead of the primary antibody. In the second step, the biotinylated rabbit anti-rat secondary antibody (DAKO) was applied. After rinsing the sections with PBS, the avidin-biotin complex (Vector Labs, Burlingame, CA, USA) was added for 30 min. Freshly prepared AEC development buffer (50 mM acetate buffer, pH 5.0), AEC solution (Sigma-Aldrich, Munich, Germany), 30% H_2_O_2_) was pipetted onto the sections. The gradual red staining of mouse endothelial cells was followed under the microscope and stopped after 10 min by adding distilled water. Counterstaining with hematoxylin for 10 min was performed to better assess tissue structure. To determine the number of vessels in the tumors, 3 to 4 field-of-view images were taken per tumor preparation using a bright-field microscope at 200× objective magnification. In each field of view, the red-stained vessels were counted using ImageJ v145.

### 2.10. Ki67 Histology

To determine the fraction of proliferating tumor cells, immunohistochemical staining was done with an antibody against human Ki67 (M7240, DAKO). 10 µm-thick tumor cryosections were fixed with a 4% PFA solution. Further treatment of the slides was performed according to the protocol already described for CD31 staining. The anti-human Ki67 antibody was applied to the sections at a concentration of 0.0016 mg/mL. The primary antibody was pretreated with a biotinylation reagent (Animal Research Kit, DAKO) for 20 min. In addition, the primary antibody was mixed with a blocking reagent (DAKO) for 20 min before administration to the cryosections. After incubation overnight at 4 °C, the avidin-biotin complex was applied to the tumor tissue sections for 30 min, which was later developed with AEC. The applied staining reaction was stopped after complete red staining of the proliferating Ki67 containing tumor cells using distilled water. Counterstaining with hematoxylin was used to improve the identification of the non-proliferating tumor cells. Counting of Ki67-labeled cells in 3 to 4 fields of view imaged at 200× magnification was performed using the ImageJ v145. 

## 3. Results

### 3.1. Expression of Chemerin, CMKLR1, GPR1, and CCLR2 in CRC and Other Cancer Cell Lines

To characterize the expression of the primary molecules involved in chemerin signaling in colorectal carcinoma (CRC) and other cancer cell lines, RT-qPCR was performed on cDNAs from such cells. Five CRC and 35 other cell lines, including a CMKLR1-overexpressing HEK293 cell clone, were tested for mRNA levels of CMKLR1, GPR1, CCRL2, and chemerin (RARRES2). High levels of CMKLR1 were only found in the transfected HEK293 clone and the breast cancer line DU4475; CRC cell lines showed moderate expression of CMKLR1. In contrast, higher GPR1 levels were found in a large number of cell lines, including the CRC cell lines LS174T and SW620 as well as in the neuroendocrine colon tumor cell line LCC18. CCRL2 was highly expressed in individual tumor cell lines from the brain, stomach, pancreas, and breast. The chemerin gene RARRES2 showed high expression in quite a number of tumor cell lines, e.g., from the brain and pancreas. CRC cell lines HT29 and SW620 also demonstrated substantial RARRES2 mRNA expression (Figure 1).

### 3.2. Secretion of Chemerin from Colorectal and Other Cancer Cell Lines

To verify whether mRNA expression of RARRES2 correlates with the amount of secreted chemerin, supernatants from 33 cell lines were analyzed after 24 h of culture using a chemerin-specific ELISA. Secreted chemerin levels above 0.5 ng/mL were found in two CRC cell lines (HT29, SW620) and two glioblastoma cell lines (N31, N39). Most of the tumor cell lines, however, showed low or no levels of secreted chemerin (Figure 2).

After having established mRNA levels for chemerin and its receptors as well as secreted amounts of chemerin in culture, the functional relevance of these expression patterns in CRC cells was to be studied. Three CRC cell lines with differential expression properties were chosen for further analysis. HCT116 shows moderate to low expression of CMKLR1, GPR1, and chemerin, HT29 is characterized by low to moderate levels of all three receptors but substantial chemerin secretion, and SW620 expresses higher levels of GPR1, moderate to low levels of the other two receptors, but releases high amounts of chemerin peptide. These three cell lines were investigated for a potential influence of chemerin signaling on three relevant tumor cell characteristics: proliferation, colony formation, and migration. As a continuation in an animal model was anticipated, luciferase-transduced cells (HCT116-luc, HT29-luc, and SW620-luc) were used in these in vitro experiments instead of the wild-type cell lines. As chemerin is known to be subject to fast proteolytic degradation, the metabolically stabilized chemerin peptide analog CG34, a potent agonist of CMKLR1 and GPR1 [31], was utilized in all following experiments. Further, the pharmacological equivalence of chemerin and its peptide analog CG34 was demonstrated in a cell based-assay (Appendix A).

### 3.3. Influence of the Chemerin Analog CG34 on In Vitro Proliferation, Colony Formation, Migration

In a first approach, the impact of the chemerin analog CG34 on the proliferation of the three CRC cell lines was studied. Two independent experimental approaches were taken: a high-content analysis quantitation after DAPI-mediated visualization of cellular nuclei (Figure 3A–D) and a real-time kinetic live-cell microscopy approach (Figure 3E–H). In the nuclei-counting experiments, medium with low concentrations of FCS (0.6–1.6%) was employed as a negative control condition, while cells incubated in a medium with 10% FCS constituted positive controls with maximum stimulation of proliferation. The chemerin analog CG34 (in low-FCS medium) only slightly stimulated growth; the difference was not statistically significant in an ANOVA analysis (Figure 3B–D). Growth of the three CRC cell lines as monitored by live-cell microscopy was not significantly stimulated or inhibited by CG34 (Figure 3E–H). 

The growth of tumor cells in a two-dimensional fashion is only a rough approximation of growth conditions in vivo. Therefore, another growth assay was applied that allows monitoring proliferation in three-dimensional structures–colonies. Indeed, in a colony formation assay, a concentration of 100 nM of CG34 was able to significantly stimulate the growth of all three CRC cell lines as compared to control (medium only). HCT116-luc cells showed a proliferation increase of 10.1% (*p* = 0.0118), HT29-luc had increased by 9.7% (*p* = 0.0124), and SW620-luc were stimulated by 16.1% (*p* = 0.0045) (Figure 4A).

Apart from proliferation, the ability of tumor cells to migrate is a crucial property influencing the course and kinetics of tumor progression. In a scratch-wound assay quantifying relative changes of the wound density, chemerin receptor stimulation by CG34 at 10 µM concentration did not change cell migration in any of the three CRC cell line models versus control conditions (Figure 4B–F).

### 3.4. Impact of CG34 on In Vivo Tumor Growth, Tumor Cell Proliferation and Vessel Density

As the in vitro analysis of three CRC cell lines had yielded indications for a potential involvement of chemerin receptor signaling in growth regulation, an in vivo study was designed to address this hypothesis in an animal model. Of the three cell models used in vitro, two were chosen for the animal study: HCT116-luc (low to moderate chemerin receptor expression, no secretion of the ligand chemerin) and HT29-luc (low to moderate CMKLR1 and GPR expression, moderate chemerin secretion). SW620-luc were excluded, as they endogenously secrete high amounts of the ligand chemerin and any effect of an exogenous chemerin analog may be obscured. As the two CRC cell lines chosen were of human origin, immunodeficient NMRI nu/nu nude mice were applied to generate subcutaneous tumor xenografts of HCT116-luc and HT29-luc cells. Animals were treated by i.p. injection daily, starting from day 8 and ending on day 28 after tumor cell inoculation. After randomization, animals received either a bolus of 50 nmol of the chemerin analog CG34 or water. Both tumor size (caliper measurement), as well as tumor bioluminescence (whole-animal imaging) as a measure of both tumor size/cell number as well as tumor cell viability, were monitored on days 1, 7, 15, 21, and 28 of the experiment (Figure 5A).

From day 15 of the experiment onward, tumor volumes in the treatment and control groups started to differ for both HCT116-luc as well as for HT29-luc xenografts, with tumor volumes in the treatments groups being higher than in the control groups. On day 28, differences in tumor volumes (mean ± SD) were found to be statistically significant for HCT116-luc xenografts in the control group 53.9 ± 52.4 mm^3^ versus 189 ± 100 mm^3^ in the CG34 treatment group (*p* = 0.0151). For HT29-luc xenografts, volumes in the control group (261 ± 147 mm^3^) were considerably lower than in the treatment group, though not statistically significant (427 ± 250 mm^3^, *p* = 0.1906) (Figure 5B, Appendix A).

Similarly, bioluminescence intensity evaluated as total flux (photons per second), started to develop differently from day 15 of the experiment for both CRC cell xenografts. Again, values were higher in the treatments groups than in the control groups. On day 28, differences in bioluminescence were found to be statistically significant for HCT116-luc xenografts: in the control group, values were 6.5 × 10^8^ ± 1.1 × 10^9^ p/s versus 6.0 × 10^9^ ± 5.0 × 10^9^ p/s in the CG34 treatment group (*p* = 0.0152). For HT29-luc xenografts, differences were also high but, due to high variance, not statistically significant: in the control group, total flux was 3.9 × 10^9^ ± 3.5 × 10^9^ p/s as opposed to 9.3 × 10^9^ ± 5.0 × 10^9^ p/s in the treatment group (*p* = 0.0649) (Figure 5C, Appendix A).

At the termination of the experiment, the sizes and weights of tumors from sacrificed animals were determined. For both volume and weight, all mean values from tumors in the CG34 treatment group were considerably higher than the control group, even though, due to high variance, the differences lacked statistical significance (Figure 5D,E).

To investigate a potential involvement of tumor vascularization in growth differences between the groups in the animal study, tissue cryosections from all tumors were prepared and stained immunohistochemically with a marker antibody for blood vessels (CD31). While vessel numbers per area were not different for HCT116-luc xenografts, in HT29-luc xenografts from the CG34 treatment group, they were higher than in the control group (Figure 6A,C).

As tumor growth is associated with the regulation of cellular proliferation, cryosections were also stained with a marker antibody for mitotic activity (Ki67). In this experiment, the percentage of Ki67-positive cells (mean ± SD) was significantly higher in HCT116-luc xenografts from the treatment group (45.8 ± 9.3) than from the control (38.4 ± 10; *p* = 0.362). In the HT29-luc groups, no significant difference was found Figure 6B,D).

## 4. Discussion

Chemerin has been demonstrated to be bound by and to activate two heptahelical plasma membrane receptors, CMKLR1 and GPR1, while a third, CCRL2, binds chemerin and is believed to pass on the ligand to the other receptors [1,2]. The expression of the three receptors has been revealed to some degree in healthy organs and tissues, where they act to transmit the various chemerin-mediated physiological functions from immune cell attraction to metabolic regulation [32]. In cancer, however, the expression of CMKLR1, GPR1, and CCRL2 has not been studied to great extent. In CRC, CMKLR1 was discovered at higher abundance in CRC tumor tissues than in margins [30]. In addition, CMKLR1 was found to correlate with GDF-15 and VEGF-A levels in CRC tumor-free margin [33]. Our own previous report revealed CMKLR1 to be highly expressed in the breast cancer cell line DU4475 [31]. The current study shows quantitative mRNA abundance data for five CRC and 35 other cell lines, revealing the complex and differential expression patterns that do not point to a particular, specific role for any of the receptors in one of the tumor entities under investigation. Rather, CMKLR1 and GPR1 expression levels vary across many orders of magnitude for most tumor types, while CCRL2 is completely missing from seven of the tested cancer cell lines (Figure 1). While RT-qPCR is a highly specific and very sensitive analytical method to quantify the presence of mRNA, it has limited value in predicting protein levels, in particular with membrane proteins. It represents, therefore, a limitation of this study, that it does not provide evidence of receptor presence at the protein level. In CRC cell lines, the three cell lines HCT116, HT29, and SW620 represent the different expression patterns and were therefore selected for subsequent experiments.

In contrast to the receptors, the ligand chemerin has been extensively characterized with regard to tumor expression and serum levels [18,19]. Most research, however, has been focusing on immune cells, cancer-associated fibroblasts, and other components of the tumor microenvironment as sources of chemerin secretion. This study demonstrated significant chemerin mRNA levels in cancer cell lines from the brain, stomach, pancreas, and colon (Figure 1). Relevant chemerin concentrations in the supernatant of cultured cells, however, were only detected in two glioblastomas as well as in three CRC cell lines, potentially suggesting a role of tumor cell-produced chemerin in the biology of these cancer types. Tumor cells as a source of chemerin so far have been studied in the context of artificial overexpression or intratumoral injection of chemerin [20,34], while secretion from cell lines has not been studied so far, to the best of our knowledge.

Several tumor types so far have been identified to feature a direct chemerin action on tumor cell proliferation and tumor growth, which may be either stimulating or inhibiting [35]. In gastric cancer, e.g., chemerin was observed to stimulate carcinogenesis by inducing phosphorylation of p38 and ERK 1/2 MAPKs. This study did not find a clear stimulatory or inhibitory action of the chemerin analog CG34 on the three CRC cell lines HCT116-luc, HT29-luc, and SW620-luc when grown in a two-dimensional monolayer. However, when these cells were monitored for three-dimensional colony formation, CG34 significantly promoted growth in all three (Figure 4A). While these results may appear contradictory, it is conceivable that the growth-enhancing effect of the chemerin analog depends on signaling only active in a three-dimensional cell context. It has been established that cells adapt to their environment by reacting to local signals, which in turn results in changes in cell proliferation and other physiological functions [36]. Still, one limitation of this study is the lack of a mechanistic explanation for the differential regulation by chemerin in 2D versus 3D cultures. Further studies will have to elucidate the signaling processes involved in chemerin-induced enhancement of CRC cell colony formation.

Similar to its action on CRC cell colonies, CG34 clearly stimulated the growth of HCT116-luc as well as HT29-luc xenografts in a subcutaneous nude mouse model, as measured by tumor volume and bioluminescence intensity (Figure 5). Statistical significance was found for differences between CG34-treated and control groups in HCT116-luc xenografts, while a high variance in values for HT29 precluded a statistically significant result. Similarly, tumor volume and weight were considerably increased. Statistical analysis found significance for the tumor volume differences between HCT116-luc cells treated with CG34 or control resulting from live measurement, but not for the corresponding values measured after the animals were sacrificed. The reason for this remains unclear and may be attributed to measurement error or imprecision. These results are indicative of a strong stimulatory function of chemerin signaling in CRC growth regulation. One of the potential influences of chemerin in regulating tumor growth that has been widely studied is its impact on the formation of new vessels [18]. Ex-vivo analysis of tumors from the animal study demonstrated only a minor, non-significant enhancement of vessel density in HT29-luc xenografts of the CG34 treatment group versus control (Figure 6A,C).

Though a correlation of chemerin levels and CRC risk has been established [27,28,29], a regulatory effect has not been reported in the literature to date. Even though this study, hence, provides the first report of a chemerin agonist stimulating CRC in an animal model, it has a number of limitations. As the chemerin analog was administered systemically via a vein, this analysis cannot distinguish direct effects of CG34 on tumor cells from indirect regulation via immune cells, cancer-associated fibroblasts, or other cells of the tumor microenvironment. Indeed, indirect effects dominate in most reports on chemerin effects in cancers [18,19,35]. It is, however, intriguing that both in in vitro colony formation as well as in the animal study, CG34 led to increased growth, potentially suggesting a direct effect of the analog on CRC cells. This is also supported by the detection of a statistically significant increase in the mitosis marker Ki67 in HCT116-luc xenografts of the CG34 treatment group versus tissues from the control group (Figure 6B,D). Further studies will have to provide confirmation as well as clues about the exact molecular nature of this tumor-promoting activity.

## 5. Conclusions

The chemerin analog CG34 proved to stimulate colony formation as well as xenograft growth in two mouse models of colorectal carcinoma, with intratumor Ki67 cells increased in one of these models. These results strongly indicate a regulating, stimulatory effect of chemerin receptor signaling in colorectal carcinoma.

## Figures and Tables

**Figure 1 cancers-14-00125-f001:**
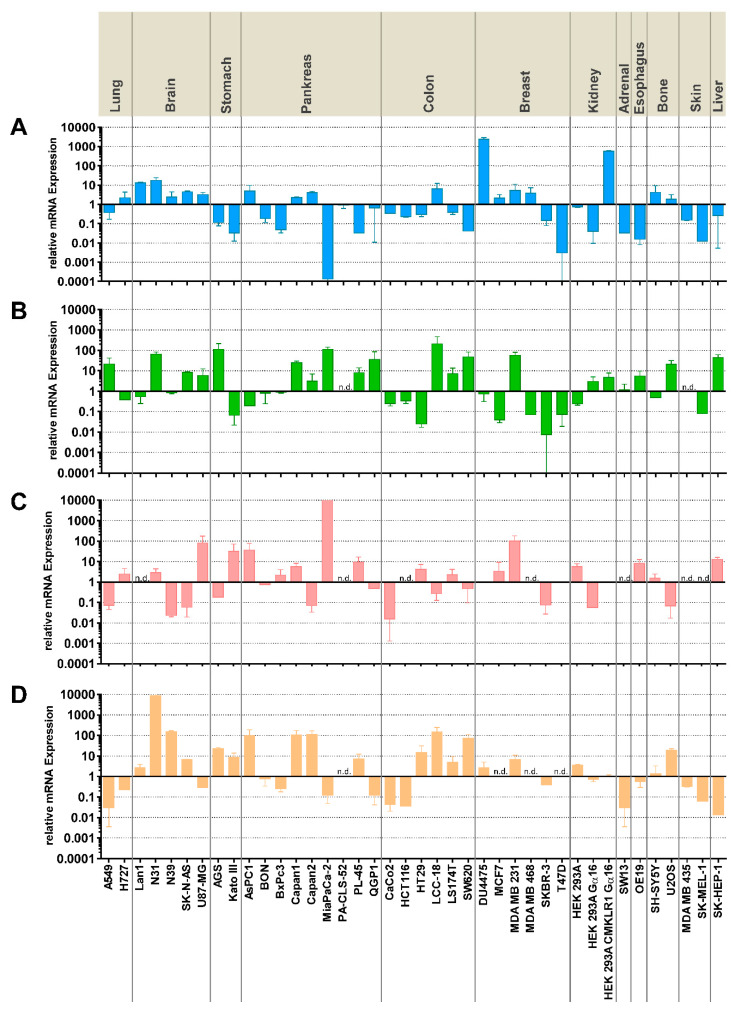
Expression of *CMKLR1, GPR1, CCRL2*, and chemerin (*RARRES2*) mRNA in human colorectal and other cancer cell lines as determined by quantitative RT-qPCR. (**A**) CMKLR1; (**B**) GPR1; (**C**) CCRL2; (**D**) chemerin (RARRES2). The upper section denotes the organ of tumor origin; cell line designations are provided at the bottom. Data have been normalized to reference genes UBC, ALG9, and GAPDH. A value of 1 in the graph corresponds to a cq of 34. All data represent mean ± S.E.M. of *n* = 2–5 independent experiments; n.d. = mRNA not detected.

**Figure 2 cancers-14-00125-f002:**
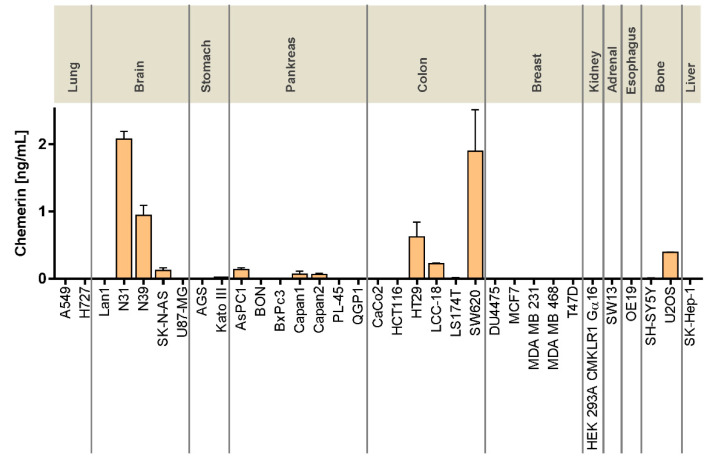
Secretion of the peptide chemerin by CRC and other cancer cell lines as determined by ELISA quantitation of cell culture supernatants. Cells had been cultured at a density of ~70%; supernatants were collected after 24 h of incubation. The upper section denotes the organ of tumor origin; cell line designations are provided at the bottom. All data represent mean ± SD of *n* = 2–6 independent experiments performed in duplicate.

**Figure 3 cancers-14-00125-f003:**
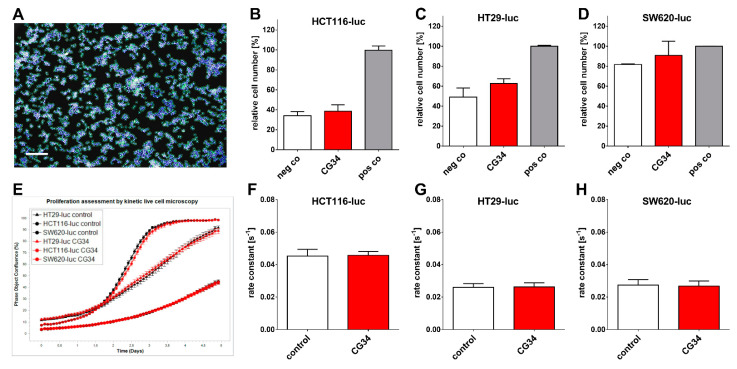
Influence of the chemerin peptide analog CG34 on the proliferation of three luciferase-expressing colorectal carcinoma cell lines (HCT116-luc, HT29-luc, SW620-luc) in vitro as determined versus negative control (low-FCS medium) and positive control (high-FCS medium) by an endpoint assay (IN Cell high-content analysis of DAPI-stained cell nuclei, upper panel **A**–**D**) and versus control (medium only) by a kinetic assay (confluence measurement using Incucyte live-cell microscopy, lower panel **E**–**H**). (**A**) Exemplary image of the pattern recognition algorithm detecting DAPI-stained nuclei (scale bar: 50 µm); quantitative results for (**B**) HCT116-luc; (**C**) HT29-luc; (**D**) SW620-luc. (**E**) Exemplary result of a kinetic live-cell microscopy experiment over five days (black: control cells; red: CG34-treated cells). Quantitative results for (**F**) HCT116-luc; (**G**) HT29-luc; (**H**) SW620-luc. All data represent mean ± SD of *n* = 3 (DAPI cell nuclei counting) or *n* = 5 (kinetic microscopy) independent experiments.

**Figure 4 cancers-14-00125-f004:**
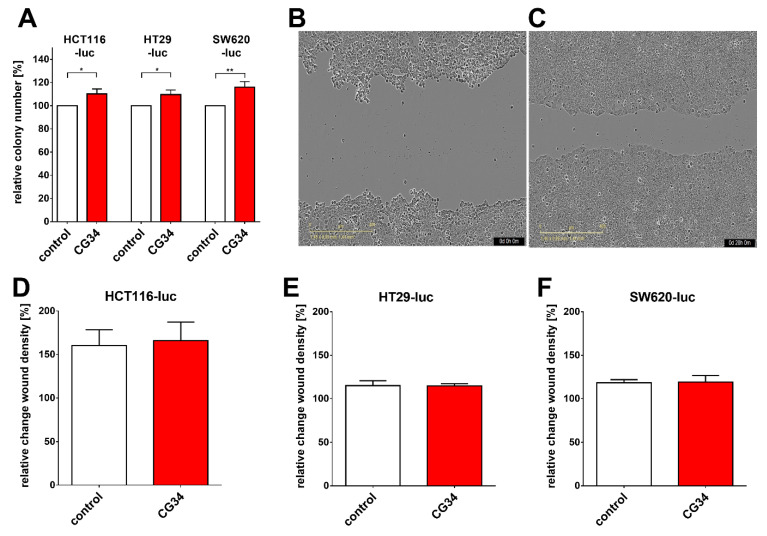
Influence of the chemerin peptide analog CG34 versus control (medium only) on colony formation and migration of three luciferase-expressing colorectal carcinoma cell lines (HCT116-luc, HT29-luc, SW620-luc). (**A**) Results of the colony formation assay. Colony formation data represent mean ± SD of *n* = 3–4 independent experiments performed in triplicate. Statistical significance was determined with an unpaired *t*-test; * *p* < 0.05, ** *p* < 0.01. (**B**) Exemplary image of a migration experiment at the start of treatment (0 h). (**C**) Image of the same well as in (**B**), taken 20 h later. (**D**–**F**) quantitative results of the migration assay: (**D**) HCT116-luc; (**E**) HT29-luc; (**F**) SW620-luc. Migration data represent mean ± SD of *n* = 4 independent experiments performed in 9-tuplicate.

**Figure 5 cancers-14-00125-f005:**
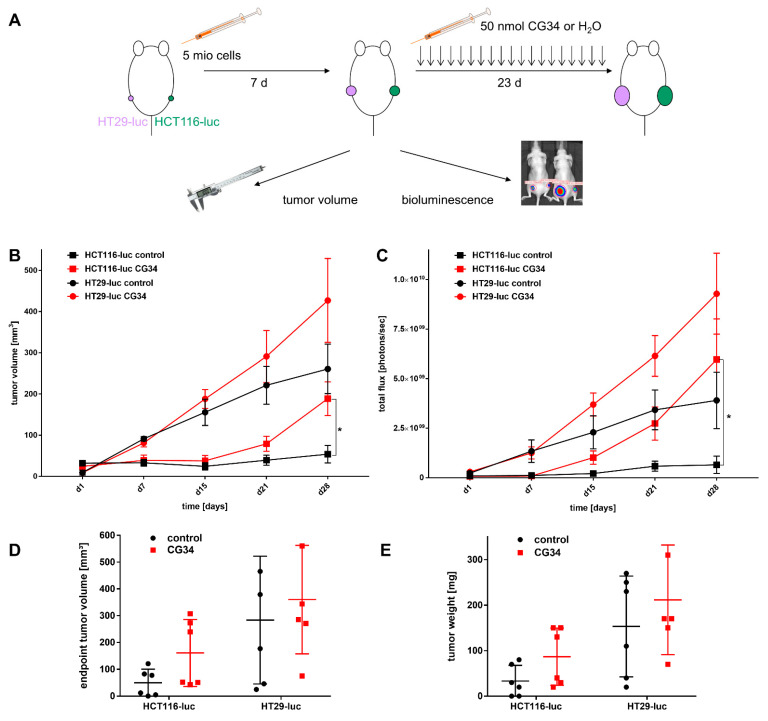
Influence of the chemerin peptide analog CG34 on the growth of two human CRC xenograft mouse models. (**A**) Schematic representation of the tumor inoculation and treatment process. Five million of the luciferase-expressing HCT116-luc and HT29-luc cells were used to inoculate female NMRI nu/nu mice. On day 8 after inoculation, daily treatment with i.p. injections of CG34 or water was started in an investigator-blinded fashion; treatment was continued until day 28. Tumor volume and bioluminescence were recorded weekly. (**B**) Tumor volume measurements performed with a caliper; (**C**) Bioluminescence measurements performed with a high-sensitivity CCD imager; (**D**) Volume measurements performed with a caliper at the end of the experiment after mice had been sacrificed; (**E**) Tumor weight measurements performed at the end of the experiment using explanted tumors. Volume and bioluminescence data (**B**,**C**) represent mean ± S.E.M. of *n* = 6 tumors per group. Statistical verification was performed using an unpaired *t*-test or a Mann-Whitney test; * *p* < 0.05. Endpoint tumor volume and tumor weight data (**D**,**E**) represent mean ± SD of *n* = 6 tumors per group.

**Figure 6 cancers-14-00125-f006:**
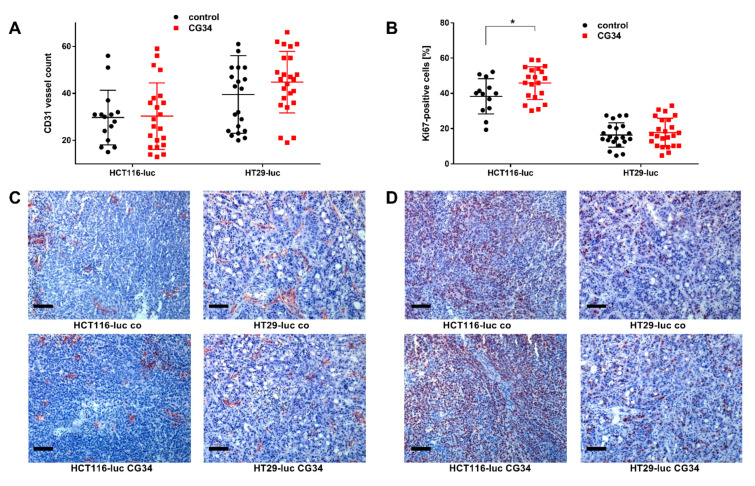
Immunohistological analysis and quantitation of vessel density (CD31 staining; (**A**,**C**)) and proliferative activity (Ki67 staining; (**B**,**D**)) in sections of xenograft tumors after the treatment study (Figure 5). Quantitative data (**A**,**B**) represent mean ± SD of *n* = 15–24 analyzed microscopic imaging fields. Statistical testing was performed using an unpaired *t*-test; * *p* < 0.05. Scale bars: 50 µm.

**Table 1 cancers-14-00125-t001:** Oligonucleotide primer information (fwd: forward, rev: reverse).

Target Gene	Source	Sequence, If Not Available: Product Number
*CMKLR1*	Roche	137620
*GPR1*	SIGMA	fwd: CTGGAACCGGGAAGGTACAC rev: TCCCAGCTGGACTTTCTCCT
*CCRL2*	Life Technologies	Hs00277231_m1
*RARRES2*	Life Technologies	Hs00161209_g1
*ALG1*	SIGMA	fwd: GTCTTCTGGCTTTTGTGAGCTG rev: TCACGTGCAACCCAAACTTC
*GAPDH*	Life Technologies	4326317E-1105051
*UBC*	SIGMA	fwd: ATTTGGGTCGCGGTTCTTG rev: TGCCTTGACATTCTCGATGGT

## Data Availability

Numerical, machine-readable data for this study have been deposited in an open data repository for public access: http://doi.org/10.5281/zenodo.5710900 (accessed on 23 December 2021).

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
