# Peer review of "A Chemerin Peptide Analog Stimulates Tumor Growth in Two Xenograft Mouse Models of Human Colorectal Carcinoma"

_cancers, 2021, doi:10.3390/cancers14010125_

Round 1

Reviewer 1 Report

In this study, Friebus-Kardash et al. investigate the effects of a chemerin peptide on human colorectal cancer cell growth. ProfilingRARRES2 (chemerin) transcripts and those encoding its receptors (CMKLR1,GPR1and CCRL2) identified several established cancer cell lines, including colorectal cancer lines, that expressed high levels of ligand and receptors. The CRC lines HCT116 and SW620 were found to secrete chemerin peptide into the media. Whereas treatment of cells with the chemerin analog CG34 did not affect growth on plastic, it did stimulate growth when assessed in a 3D colony forming assay. CG34 did not affect migration in a scratch wound assay. In a xenograft tumor assay, CG34 treatment significantly augmented growth of HCT116 tumors and also stimulated growth of HT29 tumors, but results in the latter line did not achieve statistical significance. Explanted tumors did not display increased vascularization, but HCT116 tumors did contain increased numbers of Ki67+cells.

This is a nice study that advances our understanding of the pro-tumorigenic role of chemerin in CRC. A few points should be addressed for clarity and these are listed below:

  1. Primer sequences should be included in the materials and methods section for the genes whose expression were profiled in Figure 1.

  1. Figure 5 would benefit from a diagram of the tumorigenesis and treatment strategy.

  1. In Figure 5, can the authors explain why the tumor volume data for HCT116-luc was not significant relative to control in panel C, but was significant at day 28 in panel A? In addition, for this experiment, it would be helpful if an image depicting the primary explanted tumors for the control and treated groups was provided. This would help to clarify whether one or two of the tumors of the six, especially for the HT29-luc group, is driving the variance in tumor volume.

  1. The images in Figure 6C and D require improvement. In C, the CD31 staining is not clear and in D, there appears to be many more Ki67+ cells in the fields provided for HCT116 than quantified in the bar graph (panel B). Moreover, the Ki67+cells for treated HT29-luc tumors are not readily discernable. Higher magnification of representative fields used for quantification is required.

Author Response

  1. Primer sequences should be included in the materials and methods section for the genes whose expression were profiled in Figure 1.

Reply: We thank the reviewer for pointing out our omission. We added a table (Table 1) that now contains the sequences of forward and reverse primers. For Taqman primer sets, the company considers primer sequences proprietary. In these cases, we have provided product numbers to allow readers replication of the experimental conditions used in this study.

  1. Figure 5 would benefit from a diagram of the tumorigenesis and treatment strategy.

 Reply: We have added a diagram that depicts the tumor initiation and treatment process.

  1. In Figure 5, can the authors explain why the tumor volume data for HCT116-luc was not significant relative to control in panel C, but was significant at day 28 in panel A? In addition, for this experiment, it would be helpful if an image depicting the primary explanted tumors for the control and treated groups was provided. This would help to clarify whether one or two of the tumors of the six, especially for the HT29-luc group, is driving the variance in tumor volume.

Reply: We thank the reviewer for this comment. Indeed, our statistical analysis found significance for the tumor volume differences between HCT116-luc cells treated with CG34 or control, but not for the corresponding values measured after the animals were sacrificed. The reason for this is unclear and may be due to measurement error or imprecision. We have added this in the discussion.

We agree with the reviewer that images of the individual tumors would be helpful to illustrate differences between the two groups. Unfortunately, such images were not taken at that time. In order to visualize the different individual tumor sizes, we have added a new Supplementary Figure 2 that shows these values, grouped by tumor type and treatment group. We have added a similar graph for the bioluminescence data (Supplementary Figure 3). We have, however, decided to keep the figures for tumor volume and bioluminescence in the main part of the manuscript as they are, because they provide a better overview of the tumor growth kinetics and treatment influence. Furthermore, the bar graphs for tumor endpoint volume and tumor weight data were replaced by scatter plots to better visualize the distribution of individual values (Figure 5 D, E).

In addition, readers will be able to obtain individual numerical data from the data set we have uploaded and for which we have provided the link (http://doi.org/10.5281/zenodo.5710900) in the manuscript’s Data Availability Statement.

  1. The images in Figure 6C and D require improvement. In C, the CD31 staining is not clear and in D, there appears to be many more Ki67+ cells in the fields provided for HCT116 than quantified in the bar graph (panel B). Moreover, the Ki67+cells for treated HT29-luc tumors are not readily discernable. Higher magnification of representative fields used for quantification is required.

Reply: We have substituted all images by higher-quality pictures of the same tissue stainings.  The authors are indebted to the reviewer for spotting a discrepancy between graph and images: the graph for Ki67 quantification in fact shows percentage of positive cells, not cell count. We have corrected our error in graph and text.

Reviewer 2 Report

In this manuscript, the authors describe the adipokine chemerin to be elevated in some types of cancer, including colorectal carcinoma, but the functional role of chemerin in this context is not clarified. The authors use the chemerin analog CG34 to analyze proliferation, colony formation and migration in three different human colorectal cancer cell lines and in addition on two xenograft mouse models of colorectal cancer. The results suggest that there is a stimulatory role of chemerin on the growth of colorectal carcinoma.

In the introduction, they start with describing the general role of chemerin before they explain the context of chemerin in cancer in more detail. Here, they point out the contrasting discoveries of various papers of both antitumor activity and tumor-promoting effects. This allows a better understanding of the aim to analyze the role of chemerin in colorectal carcinoma. All information are fully referenced.

In addition it is suggested in line 44 “It exerts its action via its receptors CMKLR1, GPR1 and CCRL2” to add the fact that the receptors are G protein-coupled receptors and to use the full name of the receptor once before using the abbreviation.

The material and method part is explained in great detail and enables to reproduce the experiments.

First, mRNA expression of chemerin and all three receptors were tested on various cell lines from different tissues, as well as the amount of secreted chemerin was analyzed. Three colorectal carcinoma cell lines with differential expression properties were chosen for further analysis. Here, in addition to mRNA amount, the expression of the receptors on the chosen cells should be analyzed because mRNA amount and protein amount does not always automatically match.

They further explain that they use the metabolically stabilized chemerin peptide analog CG34 to analyze proliferation, colony formation and migration. The first analysis of proliferation is shown in figure 3. Readers should be able to read figures on their own, therefore the information on neg co and pos co in B-D should be added in the caption. It is also not clear what control means in F-H. The font in 3E is too small to be read.

Next, the authors tested the growth of colonies. Is should be mentioned that the results are shown in 4A. In neither the results part nor in figure 4, it is mentioned what the control condition for the experiment has been used, this has to be added.

In general, all experiments are planned and performed very well. In their discussion part they correctly state, that a complete transfer of the data from the peptide analog to full length protein is not possible. To enhance the relevance of the manuscript and to get further knowledge about the influence of chemerin in colorectal carcinoma, it is suggested to perform the most promising cell based assays also with the full-length protein. To avoid loss of protein due to degradation over time, media with protein should be exchanged or newly added e.g. every 24 hours as it was already done in the experiments with the peptide analog. Then the data would offer a certain added value and would also be better suited to the actual purpose of the work to analyze the role of chemerin in colorectal cancer.

Author Response

In addition it is suggested in line 44 “It exerts its action via its receptors CMKLR1, GPR1 and CCRL2” to add the fact that the receptors are G protein-coupled receptors and to use the full name of the receptor once before using the abbreviation.

Reply: We thank the reviewer for this comment and have now added the full designations of the three receptors as well as the fact that these are G protein-coupled receptors.

Here, in addition to mRNA amount, the expression of the receptors on the chosen cells should be analyzed because mRNA amount and protein amount does not always automatically match.

Reply: The authors fully agree with the reviewer in that a linear correlation between mRNA amount and presence of protein cannot be readily assumed. Unfortunately, none of our attempts at demonstrating receptor protein in either immunofluorescence or Western blotting, using multiple antibodies, yielded acceptable and reliable results. Quite a number of studies have demonstrated that specific antibodies against G protein-coupled receptors can be difficult to obtain, most likely because the immersion of the heptahelical protein in the lipid membrane and its flexibility prevent the formation of high-affinity antibodies. Most commercially available GPCR antibodies were of limited values in our hands. So one of the reasons we did not detect the receptors probably was a lack of suitable antibodies. In addition, according to the Ct values of the RT-qPCR experiments, we assume receptors to be present in all three cells, yet at low to moderate densities. In order to address this important consideration and to point out this limitation of the study, we have added a short paragraph to the discussion, where we discuss the lack of protein detection versus positive mRNA detection in these cells.

The first analysis of proliferation is shown in figure 3. Readers should be able to read figures on their own, therefore the information on neg co and pos co in B-D should be added in the caption. It is also not clear what control means in F-H. The font in 3E is too small to be read.

Reply: The authors thank the reviewer for these helpful comments. We have now added the nature of the controls condition (low-FCS medium, high-FCS medium, medium only) in the figure caption for B-D and F-H. In 3E, we have increased the font size of the figure description.

Next, the authors tested the growth of colonies. It should be mentioned that the results are shown in 4A. In neither the results part nor in figure 4, it is mentioned what the control condition for the experiment has been used, this has to be added.

Reply: We thank the reviewer for pointing out this omission. We have now added the reference of Figure 4A and the description of the control condition (medium only) in the corresponding paragraph as well as in the figure legend.

It is suggested to perform the most promising cell based assays also with the full-length protein. To avoid loss of protein due to degradation over time, medium with protein should be exchanged or newly added e.g. every 24 hours as it was already done in the experiments with the peptide analog. Then the data would offer a certain added value and would also be better suited to the actual purpose of the work to analyze the role of chemerin in colorectal cancer.

Reply: The authors thank the reviewer for these remarks regarding the pharmacological equivalence of CG34 versus the full-length chemerin protein, which is of high significance to the results described. In order for two chemical entities to be pharmacologically equivalent, they should have equal potency at their receptors. In the supplement, we now provide method and result of a cell-based signaling assay demonstrating quantitative functional pharmacological equivalence of CG34 and full-length chemerin protein. In addition, we have also performed a screening for CG34 binding to 118 other GPCRs and further drug targets such as ion channels, with none of them showing significant binding to CG34 (data not shown). We conclude from this screening experiment, that CG34 has the same binding profile as chemerin. Consequently, the results obtained in this study with the analogue CG34 can be regarded representative of the same pharmacological activity as would be seen with the authentic natural chemerin protein.

Round 2

Reviewer 1 Report

The authors did a nice job addressing all concerns I had with the previous version.